# Deformity Reconstruction Surgery for Blount’s Disease

**DOI:** 10.3390/children8070566

**Published:** 2021-06-30

**Authors:** Craig A. Robbins

**Affiliations:** Paley Orthopedic & Spine Institute, West Palm Beach, FL 33407, USA; crobbins@paleyinstitute.org; Tel.: +1-5618445255

**Keywords:** Blount’s disease, infantile, early-onset, late-onset, juvenile, adolescent, tibia vara

## Abstract

Blount’s disease is an idiopathic developmental abnormality affecting the medial proximal tibia physis resulting in a multi-planar deformity with pronounced tibia varus. A single cause is unknown, and it is currently thought to result from a multifactorial combination of hereditary, mechanical, and developmental factors. Relationships with vitamin D deficiency, early walking, and obesity have been documented. Regardless of the etiology, the clinical and radiographic findings are consistent within the two main groups. Early-onset Blount’s disease is often bilateral and affects children in the first few years of life. Late-onset Blount’s disease is often unilateral and can be sub-categorized as juvenile tibia vara (ages 4–10), and adolescent tibia vara (ages 11 and older). Early-onset Blount’s disease progresses to more severe deformities, including depression of the medial tibial plateau. Additional deformities in both groups include proximal tibial procurvatum, internal tibial torsion, and limb length discrepancy. Compensatory deformities in the distal femur and distal tibia may occur. When non-operative treatment fails the deformities progress through skeletal maturity and can result in pain, gait abnormalities, premature medial compartment knee arthritis, and limb length discrepancy. Surgical options depend on the patient’s age, weight, extent of physeal involvement, severity, and number of deformities. They include growth modulation procedures such as guided growth for gradual correction with hemi-epiphysiodesis and physeal closure to prevent recurrence and equalize limb lengths, physeal bar resection, physeal distraction, osteotomies with acute correction and stabilization, gradual correction with multi-planar dynamic external fixation, and various combinations of all modalities. The goals of surgery are to restore normal joint and limb alignment, equalize limb lengths at skeletal maturity, and prevent recurrence. The purpose of this literature review is to delineate basic concepts and reconstructive surgical treatment strategies for patients with Blount’s disease.

## 1. Introduction

Blount’s disease is eponymous with non-physiologic tibia vara in skeletally immature patients. It is an idiopathic developmental abnormality affecting the medial proximal tibia growth plate, resulting in a multiplanar deformity with pronounced tibia varus [1,2]. The first clinical and radiographic description of a single patient was by Erlacher in the German literature in 1922 [3]. In 1937, Blount provided detailed histologic and radiographic descriptions of 13 original cases of tibia vara in the English literature [4]. He coined the term “osteochondrosis deformans tibiae” and characterized the early-onset form (herein referred to as infantile tibia vara (ITV)) that is clinically apparent before age four and the late-onset form (herein referred to as late onset tibia vara (LOTV)) that develops in older children prior to skeletal maturity [5]. In 1990, Thompson and Carter re-classified Blount’s disease into three age-onset groups: (1) infantile: through three years of age, (2) juvenile: four to ten years of age, and (3) adolescent: 11 years or older before skeletal maturity [6]. The juvenile form is the least common and exists with characteristics of the infantile and late-onset forms. This discussion will utilize ITV and LOTV as the main categories of Blount’s disease.

ITV is often bilateral, is more likely in females, and has more severe tibial deformity than LOTV [7] (Figure 1). LOTV is more often unilateral and more often has additional femoral deformities [8]. Both are more prevalent in Blacks, Hispanics, and Scandinavians than other populations and nationalities, especially LOTV [9,10]. The tibia vara is the predominant feature in limbs affected by Blount’s disease and, if left untreated, progresses to knee deformity, gait abnormalities, and premature medial compartment knee arthritis [11,12,13].

## 2. Etiology and Pathophysiology

A single cause for Blount’s disease is unknown, and it is currently thought to result from a multifactorial combination of genetic, humoral, biomechanical, and developmental factors [14,15,16,17]. Early walking and obesity suggest a mechanical contribution to deformity in ITV [18]. LOTV is also linked with obesity [6,19]. A relationship with vitamin D deficiency exists in all types [19,20,21,22,23]. Wenger et al. proposed that age-related differences in osseous physiology lead to the clinically and radiologically distinct entities of early- and late-onset disease [24]. In young children, the tibial ossification center is cartilaginous and pliable; in adolescence, however, the tibial epiphysis is well-formed and bony and does not deform. Repetitive compressive forces across the medial knee in a predisposed young child with physiologic varus leads to a vicious cycle of growth inhibition with worsening varus, delayed ossification, and deformity with characteristic sloping of the medial epiphysis [24]. Wenger et al. postulated that most cases of LOTV occur in children with mild residual physiologic genu varus. With the adolescent growth spurt and rapid weight gain, the borderline varus suddenly worsens due to a similar mechanism of pressure-induced growth-inhibition of the medial physis [24].

Regardless of the etiology, the clinical and radiographic findings are consistent with each type of Blount’s disease, and the severity of deformities relates to the age of onset and therefore the duration and amount of growth suppression of the medial proximal tibia physis. ITV is characterized by more severe deformity of the proximal tibia with medial plateau depression. Despite the medial growth disturbance, the lateral tibial physis and fibula grow normally. Because the fibula is slightly inclined in the sagittal plane (more posterior proximal and more anterior distal), relative overgrowth creates a crank-shaft phenomenon resulting in internal tibial torsion [5]. Schoeneker et al. reported a significant distal femoral valgus in four of seven children with advanced ITV [25]. Because they begin at a later age, the deformities in LOTV are often less pronounced, and there is not the medial tibial plateau depression seen with ITV (Figure 2). Distal femoral varus is common, and Sabharwal reported that it contributes up to one third of the total varus deformity in LOTV [7,8,9,10,11,12,13,14,15,16,17,18,19,20,21,22,23,24,25,26,27]. Increased femoral anteversion has been reported as well [28]. As the knee deformities progress in both types, significant strain is put on the lateral collateral ligament, which can lead to laxity, joint line divergence, and knee instability [29]. The pathologic changes in Blount’s disease progress over time to develop complex multiplanar deformities often with additional deformities in the same limb.

## 3. ITV Differential Diagnosis and Clinical Features

Bow-leg deformity in infants and children is common. The differential diagnosis includes metabolic conditions such as Rickets and renal osteodystrophy, congenital deformity such as tibial hemimelia, genetic conditions such as bone dysplasias, post-infectious and post-traumatic etiologies, physiologic bowing, and ITV [30]. A comprehensive history and thorough physical exam limit the differential list; genetic and metabolic abnormalities often involve additional organ systems and can affect the entire skeletal system, and the history suggests acquired causes for deformity. The clinical exam should include strength, rotational profile, joint range of motion, joint stability, neuromuscular function, limb length measurement, gait pattern, and spine alignment.

Physiologic bowing and ITV are often difficult to distinguish from each other. The combination of hip flexion and external rotation with increased internal tibial torsion worsen the appearance of genu varus with knee flexion in children with physiologic bowing. The salient clinical features in ITV are a painless varus deformity of the knee with internal tibial torsion. In unilateral cases there is a leg length discrepancy. There is a characteristic lateral thrust with gait, and knee stability testing may show lateral ligamentous laxity [15]. Often there is a palpable prominence on the medial proximal tibia metaphysis corresponding with the site of pathology. In general, ITV has a more abrupt deformity centered on the proximal tibia, whereas physiologic bowing shows a gentler curve affecting the entire leg below the knee.

The Cover Up test described by Davids et al. is an accurate clinical screening tool to differentiate between physiologic bowing and ITV [31]. It is a qualitative test comparing the clinical alignment of the distal thigh to the top portion of the leg with the child supine and the legs rotated so the patellae are forward. The middle portions of the thighs and legs are covered up so only the knee is visible. Neutral or varus alignment at the lateral knee is considered a positive test (Figure 3) and suggests the possibility of ITV. Obvious valgus appearance at the lateral knee is considered negative (Figure 4) and indicates physiologic bowing. In their study, all patients with radiographic ITV had a positive test (sensitivity = 1.00), and 18 of 25 children with a positive test had or developed ITV (positive predictive value = 0.72). A total of 43 of 50 children with physiologic bowing had a negative test (specificity = 0.86), and all children with a negative test had physiologic bowing (negative predictive value = 1.00). Radiographic evaluation and routine follow-up are warranted in any child with a positive test or suspicion of ITV.

## 4. Radiographic Imaging

A standard X-ray series at our institute included full-length standing frontal and lateral radiographs. For the AP view, the patellas are rotated in a forward position (Figure 5), the beam is centered at the knee, and blocks are used under a shorter limb to level the pelvis. In large patients it is often necessary to image each leg individually in the AP view. Weight-bearing films may demonstrate knee joint laxity with widening of the lateral joint space. Standing AP and lateral images centered on the knee on cassettes long enough to include the entire tibia are taken as well. CT and MRI can be useful to define the anatomy of joint surfaces and physes. Rotation profile CT scans can quantify torsional abnormalities above and below the knee.

## 5. Radiographic Analysis

A line drawn from the center of the femoral head to the center of the ankle defines the load-bearing mechanical axis of the lower extremity (Figure 6). This axis is superior to the historic tibial–femoral shaft angle (Figure 7) described by Salenius and Vannka [33] for evaluation of lower extremity alignment [34]. Radiographic deformity analysis begins with the malalignment test to identify the mechanical axis deviation. Joint orientation angles are measured to identify the locations and quantify angular deformities based on population norms [32,35]. Standard coronal plane measurements include the mechanical axis deviation (MAD), femoro–tibial angle, neck shaft angle (NSA), mechanical lateral distal femoral angle (mLDFA), knee joint line congruency angle (JLCA), medial proximal tibial angle (MPTA), tibial metaphyseal–diaphyseal angle (MDA) if very young, medial tibial plateau depression angle (if present), and lateral distal tibial angle (LDTA). Standard sagittal plane measurements include the anatomic posterior distal femoral angle (aPDFA), anatomic posterior proximal tibial angle (aPPTA), and the anatomic posterior distal tibial angle (aPDTA) [36].

## 6. Diagnosis

Radiographic imaging can be used to diagnose ITV and help distinguish it from physiologic bowing, and metabolic, genetic, and acquired causes such as those due to trauma and infection in young children. A metabolic work-up may be indicated to evaluate for causative factors such a vitamin deficiency. True physiologic bowing is characterized by a varus mechanical alignment with otherwise normal appearing standing X-rays. ITV has a spectrum of characteristic radiographic changes (Figure 8) including radiolucency, sclerosis, apparent fragmentation, and angulation of the medial proximal tibia [4,38]. Levine and Drennan measured the metaphyseal diaphyseal angle (MDA) (Figure 9) of the proximal tibia in standing radiographs and concluded that patients with angles ≥ 11° were more likely to progress to ITV [39]. Feldman and Schoenecker reported that MDA angles ≤ 9° suggest physiologic varus, and angles ≥ 16° are indicative of ITV [37].

## 7. Staging

Langenskiöld described a radiographic classification (Figure 10) of changes to the medial proximal tibia epiphysis, physis, and metaphysis in ITV [14,38,40]. Deformity progresses with age through six stages culminating with complete closure of the medial physis and significant deformity. Although significant inter-observer variability has been reported, particularly in the middle stages (III–IV), the Langenskiöld classification system provides the clinician with a prognostic tool and general guidelines for treatment. In general, the earlier stages (I–III) have more potential for spontaneous resolution, and the latter stages (IV–VI) have more potential for deformity progression [41,42,43]. MRI findings correlate with radiographs, but the expense and need for sedation in toddlers make this impractical for most diagnostic uses [44].

## 8. Natural History

The natural clinical profile of physiologic bowing (Figure 11) is maximal varus around 16 months moving into valgus by age 3 to 4 years and then neutral alignment achieved around age 6–8 years of age [33]. The natural history of ITV is towards deformity progression in the majority of affected children, but spontaneous resolution has been reported [43]. As varus deformity increases, the mechanical axis moves further from midline and increases forces across the medial compartment of the knee. This can lead to development of osteoarthritis [45,46,47].

## 9. Non-Operative Treatment

In a young child with suspected ITV but no clear clinical or radiographic diagnostic criteria, a short trial of observation is warranted. Spontaneous resolution of early ITV has been described but the majority of cases progress. At the moment of any clinical or radiographic evidence of progression, brace treatment should be initiated immediately. General bracing guidelines are for children under 3 years of age with early ITV (I or II) in whom a brace can be adequately fit and regularly worn. Birch’s criteria for anti-varus long leg bracing during ambulation is for patients aged ≤ 3 years with progressive deformity, clear radiographic evidence of ITV, or lateral thrust with ambulation [15]. Loder and Johnson found a 50% brace failure in children with stage I and II infantile Blount’s disease. Their recommendations were for children between 1½–2 ½ years of age, worn during weight-bearing to decrease compressive forces, and if not corrected after one year to proceed with surgery [48]. Authors report limited success with bracing due to multiple factors including compliance and obesity [1,49,50]. Bracing is not indicated in stages III and higher as it is ineffective and delays surgery.

## 10. Surgical Planning

The most common indication for surgery in ITV is progressive deformity with failure of conservative treatment in Langenskiöld I and II. With advanced stages, immediate surgery is indicated. The goals of surgery are to restore normal joint and limb alignment, equalize limb lengths at skeletal maturity, and prevent recurrence. The reconstructive surgical plan is formulated after a comprehensive evaluation of the patient clinically, radiographically, and psychosocially. Current problems are itemized and future problems such as acquired limb length discrepancy are predicted [51]. The patient’s age, weight, stage, severity, and number of deformities may indicate which procedures have the greatest chance for success. Psychosocial factors such as reliable transportation for routine follow-up may limit which procedures are feasible. Post-operative issues must be considered as part of the preoperative plan. These include the method of surgical site stabilization, patient mobilization, hygiene and grooming, prevention of joint contractures, and ability to understand and comply with complex instructions such as external fixator care and adjustments.

Surgical options for ITV include growth modulation such as guided growth for gradual correction with hemi-epiphysiodesis and physeal closure to prevent recurrence and equalize limb lengths, physeal arrest resection, physeal distraction, osteotomies with acute correction and stabilization, osteotomies with gradual correction with external fixation, and various combinations of all modalities. To avoid under-correction, overcorrection, and creation of iatrogenic deformities, a thorough analysis of all deformities away from the proximal tibia is paramount [7]. The surgical plan should consider staging complex reconstructions to minimize and overlap surgical procedures. For example, guided growth of modest femoral deformity can be combined with acute or gradual correction of the tibia/fibula [52]. Acute correction is reserved for moderate deformities to avoid risk of soft tissue compromise or neurovascular insult with the understanding that internal fixation requires a more aggressive and invasive surgery, and postoperative adjustments are not possible. External fixation allows gradual correction and refinement of complex deformities.

## 11. Hemi-Epiphysiodesis

Hemi-epiphysiodesis, also called “guided growth”, induces angular correction at the physis through the Hueter–Volkman principal, whereby compressive forces across a physis inhibit growth, and tensile forces across a physis stimulate growth [53,54,55]. Several open and percutaneous techniques exist, including irreversible partial growth plate destruction, staples, screws, and tension bands. The later techniques are considered reversible in that growth will resume after the hardware is removed [56]. Hemi-epiphysiodesis is contraindicated in conditions that will self-correct, and the surgeon must understand the differences between physiologic and pathologic deformities [52]. Hemi-epiphysiodesis is indicated for angular correction in early-stage infantile Blount’s disease prior to pathologic medial physeal bar formation.

Over the past 15 years, lateral tension band hemi-epiphysiodesis with a two hole plate and cannulated screws (Figure 12) has become a popular method for guided growth [52]. This technique has benefits over staples and open irreversible hemi-epiphysiodesis. The implants do not loosen or migrate, a common complication with staples, but early reports found a high failure rate with small diameter titanium screws in obese patients with Blount’s disease. Larger stainless steel screws are now available [57,58,59]. Because these implants are placed outside the periosteum and do not violate the physis, they maximize their effective fulcrum and are reversible if removed. Helfing reported on 27 patients under 4 years of age who had spontaneous resolution of tibial torsion with angular correction [60]. Scott reported an 89% success rate with lateral tension-band plating on 18 limbs in 12 children with Langenskiöld I or II ITV. Burghardt et al. reviewed the Blount’s disease literature and reported the main problems with lateral tension band are under-correction and poor predictability of correction amount. However they concluded that given the low technical difficulty and morbidity, guided growth with lateral tension band plating is the gold standard of care and should be the first treatment attempted; in the event of under-correction, osteotomies should be considered the salvage procedures [58]. Bushnel commented that the surgeon must weigh the safety and simplicity of this procedure against the much more extensive but much more predicable realignment obtained with osteotomy procedures [61]. Westburty et al. concluded that the goals of hemi-epiphysiodesis in ITV and LOTV include (1) correction of deformity to avoid need for osteotomy, and (2) prevention of progression of the deformity to facilitate subsequent surgery [62]. Assan asserted that guided growth remains the best treatment for early ITV, and that the tibial osteotomy should be reserved for neglected forms or those close to skeletal maturity [63].

## 12. Physeal Arrest Resection

The growth disturbance of the proximal medial tibia in ITV behaves initially like a partial growth arrest and progresses to form a medial physeal bar and ultimately complete medial arrest [14]. MRI and CT scan can be used to accurately map the location and quantify the size of a physeal disturbance [64]. Resection of the pathologic arrest, called epiphysiolysis (Figure 13), can theoretically result in resumption of normal growth and prevent additional deformity.

Andrade reviewed 27 limbs in 24 patients aged 5–10 who underwent medial epiphysiolysis combined with valgus osteotomy. Epiphysiolysis was performed through a medial oblique incision and involved removing the pathologic Blount lesion (the medial extent of the metaphysis, physis, and epiphysis to the weight bearing portion of the medial tibial plateau) from anterior to posterior. They resected to healthy physis, interposed cranioplast over the denuded bone surfaces, and performed a closing wedge metaphyseal tibial osteotomy into an over-corrected valgus alignment. The procedure was >80% successful in resuming growth and preventing recurrence in children under 7 years of age. They recommended alternative procedures in patients who failed previous procedures or who were older than age 7 [65]. Beck reported favorable results in three patients with stage VI lesions who underwent physeal bar resection with interposition (two fat, one cranioplast) and valgus producing osteotomy of the proximal tibia and fibula [66]. Birch reported useful resumption of growth in 7 of 13 patients after resection of Stage VI lesions [15]. Loder and Johnson resected physeal bars with methylmethacrylate interposition in three children with one good, one fair, and one poor result [48].

## 13. Osteotomies with Acute Correction

Proximal tibia and fibular osteotomies are indicated for patients with ITV who have failed previous surgery or are not candidates for less invasive forms of surgery such as guided growth (Figure 14). These include patients with advanced stages (IV–VI) of pathology who have a functional medial physeal tether that would preclude correction with simple guided growth. In general, osteotomy is indicated when a rapid complete correction is desired or when it is preferable not to operate on the physis [67]. There is no consensus in the literature regarding the ideal alignment of the lower extremity after surgical reconstruction for Blount’s disease [7], but most authors recommend over-correcting ITV into valgus before age 4 to offload the medial physis [4,14,68,69,70].

The tibial osteotomy is performed below the tibial tubercle, and several options exist including opening wedge, closing wedge, dome-type, W/M serrated, and oblique. Tibial osteotomies can be open or percutaneous. Fibular osteotomies are performed under direct visualization in the proximal half to avoid peroneal nerve injury [71]. Percutaneous tibial osteotomies are safe, simple, quick, reliable, and minimally invasive [72]. Open tibial osteotomies require extensile exposure and require expert planning and carpentry to accurately correct multiplanar deformities. Closing wedges loose length, and open wedges may require bone grafting. Dome-type osteotomies make rotational correction difficult, whereas the obliquely inclined Rab-type osteotomy allows simultaneous correction of varus and internal torsion [73]. The serrated W/M osteotomy described by Khermosh and Wientraub [74] requires accurate pre-operative planning and intra-operative osteotomy carpentry to allow for correction of rotation and up to 25 degrees of angulation through an extensile approach to the tibia. Internal fixation is not needed as the metaphyseal osteotomy fragments are interlocked and are inherently stable. The limb is immobilized in plaster of Paris. Healing is rapid due to the large metaphyseal surface area.

If fibular shortening is anticipated with the correction, an oblique fibular osteotomy is performed to allow for shortening, or a diaphyseal segment is removed [36]. Care is taken to avoid injury to the peroneal nerve and its branches [71], and a prophylactic peroneal nerve decompression and fasciotomy should be considered with rotational or large angular corrections [75]. The removed fibular segment can be used as an autograft with a concomitant medial hemi-plateau elevation osteotomy. Angular and rotational deformities can be acutely corrected and stabilized with internal or external fixation. Coronal plane instability may improve after acute correction with realignment and fibular shortening [76]. Osteotomy stabilizing options range from simple casting, pins in plaster, internal fixation with plate and screws, to external fixation, or a combination. The choice of stabilization depends on the patient’s age, size, weight, and mobility.

Complications from tibia and fibular osteotomies include vascular injury, compartment syndrome, delayed union, nonunion, hardware failure, infection, recurrence, and peroneal nerve palsy. With acute correction and internal fixation under-correction, over-correction, and loss of correction can occur.

In early stages of ITV, overcorrection into 5–10° of valgus is performed to unload the medial physis. Casting and protected weight bearing further unload the medial physis and may allow spontaneous correction of the physeal insult [15]. If an initial procedure fails or the child has advanced stage disease, the reconstruction becomes more complex as the articular and proximal tibial deformity progresses (Figure 15). The surgeon must perform epiphysiodesis of the lateral proximal tibia and entire proximal fibular physis to prevent recurrence of deformity. Limb length equalization must be planned with shoe lift, contralateral epiphysiodesis, ipsilateral lengthening, or a combination. If the valgus over-correction does not spontaneously correct, a guided growth procedure can be performed at a later date to improve alignment.

Prognostic factors for failure and recurrence of deformity in ITV include age > 4, increased stage (>II), obesity, and failure to over-correct into 5–10° of valgus [77]. Maré et al. reported that significant pre-operative medial plateau depression (>60°) and post-operative knee instability with varus alignment increased the risk of recurrence [78]. Loder and Johnson reported that a single valgus osteotomy was successful in 60–75% of children under age 4, while in older children, repeat osteotomies are needed in 60–65% of cases [48]. Ferriter et al. also noted that the recurrence of deformity is higher above age 4 and occurs over time [79]. Andrade concluded that even though a discrete bony bridge cannot be demonstrated, there is significant enough growth disturbance of the medial physis in tibias with Langenskiöld III or greater, that unloading with valgus osteotomy will fail [65]. Although Rab had good short-term results with 15 months of follow-up, later studies showed a 60% recurrence rate with longer follow-up [63].

Percutaneous osteotomy with acute correction and external fixator does not allow length correction but it does allow immediate weight bearing with a minimally invasive technique. Depending on the type of fixator, alignment can be adjusted postoperatively in clinic or in the operating room. In 2000, Smith et al. performed acute correction on 23 extremities with an Orthofix fixator with overall good results [2]. In 2006, Feldman compared 14 patients with acute correction stabilized with an EBI external fixator to 18 who underwent gradual correction with a Taylor spatial frame hexapod fixator and concluded gradual correction was more accurate [80]. Current hexapod fixators allow simultaneous correction of multiplanar deformities, including length.

## 14. Hemi-Plateau Elevation Osteotomy

Depression of the medial tibial plateau can occur in the later stages of ITV when there is a functional or actual bridge across the medial physis. The deformity may appear worse on radiographs than it actually is due to the presence of unossified cartilaginous material [15]. MRI and intra-operative arthrography can be useful to define anatomy. Langenskiöld and Riska recommended hemi-plateau elevation osteotomy for late presenting ITV in 1964 [40]. Stanitski et al. did not. They evaluated 17 tibiae in 10 patients aged 3–8 with severe ITV (stage III or higher) and found no tibia had depressions on arthrograms in 11 knees or MRI in 17 knees. They reported the “empty” space seen radiographically had cartilage-density material [81]. This study has not been repeated. Several authors reported that medial plateau elevation osteotomy is an important reconstructive option in late-stage ITV [25,82,83].

The medial tibial plateau depression angle (Figure 16) can be measured and quantified with intra-operative fluoroscopy and arthrogram. The articular deformity is corrected with an oblique osteotomy originating in the axilla of the metaphyseal beak and extending into the intercondylar area (Figure 17). Increased posterior depression can be differentially corrected with rotation in the sagittal plane. Barka et al. recently described an "inside-out" technique using a gigli saw to perform the osteotomy to avoid the risk of intra-articular fracture and medial condyle displacement [84]. The open-wedge defect is bone-grafted, and the osteotomy stabilized with internal or external fixation. This osteotomy is rarely performed alone and is combined with additional procedures to acutely or gradually correct residual deformities such as tibia vara, tibial torsion, procurvatum, and limb length discrepancy. In ITV, medial plateau elevation osteotomy should be combined with lateral epiphysiodesis of the proximal tibia to complete the tibial physeal arrest and epiphysiodesis of the proximal fibula to prevent asymmetric growth and deformity recurrence (Figure 18).

## 15. Combined Osteotomies

In late-stage or neglected ITV, there is often significant depression of the medial tibial plateau in addition to procurvatum, tibial torsion, and tibia vara. Multiple authors have reported good results performing combined osteotomies and procedures including acute elevation of the medial plateau, metaphyseal osteotomy of the tibia (and fibula) for acute or gradual correction, lateral tibia epiphysiodesis, and proximal fibular epiphysiodesis (Figure 19). Gregosiewicz et al. reported on 13 cases in 10 children with advanced ITV (IV–VI) with 8-year follow-up treated with double level osteotomy with internal fixation. They performed a lateral closing wedge tibia osteotomy and used the resected bone to graft the medial tibial plateau elevation osteotomy. They reported good results in 11 of 13 patients [18]. In 2003, Accadbled et al. described a one-step treatment for ITV in four patients: percutaneous epiphysiodesis of the superolateral tibia and proximal fibula, elevation osteotomy of the medial tibial plateau, osteotomy of the fibula, and dome-shaped metaphyseal osteotomy of the tibia, followed by progressive lengthening with an Ilizarov frame [82]. They reported all patients had healed at an average of 6 months, and at follow-up (average 6 years 10 months) all had normal limb length and alignment with congruent joint surfaces.

## 16. Epiphysiodesis

Epiphysiodesis procedures are used to completely stop growth of a physis. In unilateral ITV it is most often performed on the contralateral limb at a calculated patient age to equalize limb lengths at skeletal maturity [85,86]. In the ipsilateral limb proximal tibia and fibula epiphysiodesis is combined with medial hemi-plateau elevation osteotomy in late-stage ITV to prevent recurrence and overgrowth of the proximal fibula.

Several techniques exist for epiphysiodesis. Irreversible growth plate destruction can be performed percutaneously with drills and curettes or through more invasive open techniques such as the Phemister [87]. Instrumented techniques include screws, staples, and tension bands and can be reversible especially if the physis itself is not violated. Open, more invasive techniques have a lower rate of failure at the cost of added morbidity, but all techniques report good to excellent results [88,89,90,91].

## 17. Gradual Correction with External Fixation

Dynamic external fixation allows for gradual correction of multiplanar deformities including angulation, translation, rotation, and length (Figure 20 and Figure 21). Treatment is prolonged with frequent follow-up and demands significant commitment from the patient, family, and surgeon [15]. Preoperative discussions must clearly outline the frequency of visits, ensure comprehension of fixator mechanics, and delineate the prolonged treatment time in the fixator.

Perceived advantages of gradual deformity correction with external fixator include lower risks for deep infection and neurovascular compromise than with acute correction, the ability to fine tune the correction, early weight bearing, and lengthening [92]. Disadvantages include the prolonged duration of treatment, need for frequent follow-up and radiographs, risks of pin and wire infection, fixator malfunction/hardware failure, and potential loss of correction or fracture after fixator removal. The most common complications include superficial pin infections that respond to oral antibiotics in a majority of patients. Despite these challenges, gradual correction with or without limb-lengthening is the procedure of choice for patients with severe or complex deformities not amenable to acute correction [93].

Cheraskin et al. reported on external fixator treatment in 31 patients with complex deformities resulting from early- and late-onset Blount’s disease. Despite frequent and significant complications during treatment, they had 88% achievement of treatment goals [93]. In 2003, Feldman reported that 21 of 22 patients with ITV and LOTV achieved alignment within 3°, and all had good results treated with a Taylor spatial frame. Sachs et al. concluded that LOTV correction in children with minimal LLD and without severe procurvatum or torsion (under 2 cm) did not need fibular osteotomy or fixation [36]. In patients with lateral collateral ligament laxity and limb length discrepancy, the fibula can be captured distally and pulled distally with tibial lengthening to retain the lateral collateral ligament.

## 18. Late-Onset Tibia Vara (LOTV)

LOTV presents at an older age than ITV. The deformities include tibia vara with various amounts of tibial torsion, procurvatum, limb length discrepancy, and sagittal plane knee instability indicative of lateral collateral ligament laxity. The differential diagnosis is similar to ITV, but in most cases the diagnosis is easily made based on the history, clinical exam, and radiographs. LOTV has a different radiographic appearance than ITV; there is often widening of the medial tibial physis, and ipsilateral deformity of the distal femur and tibia are common [40]. Medial tibial plateau depression is not present. There is no evidence for bony bridge in LOTV and therefore no indication for physeal bar resection [24]. The patients are more often obese, the deformity is more often unilateral, and knee pain is more common (Figure 22). Ipsilateral femoral varus can contribute substantially to the overall varus alignment of the knee. Patients with compensatory distal tibia valgus can develop gastrocsoleus and subtalar joint contractures, which must be assessed preoperatively [26,29]. Rotation profile CT scans can quantify torsional abnormalities above and below the knee. Standard deformity imaging and analysis as described for ITV is performed.

The natural history of LOTV is unclear, and degenerative medial knee joint arthritis in the third decade is common in untreated patients [13,26,94]. There is no documented role for non-operative treatment in LOTV, and surgery is the definitive treatment. The goals are to restore knee joint orientation and a neutral mechanical limb alignment, equalize limb lengths at skeletal maturity, and prevent recurrence. In obese patients, neutral alignment often gives the appearance of valgus due to fat-thigh gait, as described by Davids [68]. The patient’s large thighs necessitate walking and standing in hip abduction, which increases the inter-malleolar distance at the ankles and appearance of genu valgus. These patients often have additional deformities of the distal femur (most commonly varus) and/or tibia (most commonly valgus), and these must be independently evaluated for surgical correction via guided growth or osteotomy to avoid over-correcting one deformity to treat another [29]. There is no consensus on how much knee joint obliquity may predispose one to arthritis due to translational and shear forces, but it is agreed that over correcting the tibia to compensate for some amount of femur varus deformity is not recommended [29]. Cooke et al. reported on an analysis of adult bowlegged patients with osteoarthritis and found the pattern of distal femur valgus and proximal tibial varus will generate lateral subluxing forces and overload the medial compartment [95]. Gordon et al. recommended treating compensatory varus of the distal femur if mLDFA was >95° and distal tibia valgus if LDTA ≤ 86° [29]. De Pablos suggested guided growth for distal femoral varus >10° in skeletally immature patients [96].

The surgical options for LOTV include guided growth, physeal distraction, acute osteotomy, gradual correction, contralateral epiphysiodesis, and combined strategies. The treating surgeon must consider comorbidities of sleep apnea and dietary deficiencies [97,98]. There is no consensus in the literature regarding the ideal alignment of the lower extremity after surgical reconstruction for Blount’s disease [7], but most authors recommend correcting to neutral mechanical alignment for LOTV [4,14,68,69,70]. The use of custom 3D printed osteotomy guides has been reported [99].

As with ITV, guided growth with lateral tension-band hemi-epiphysiodesis is a first-line surgical option for skeletally immature patients with LOTV [58]. Despite reports of inadequate correction, hardware failure, and the need for subsequent procedures, the relatively low morbidity and potential improvement of deformity make this a safe and recommended first-line procedure [57,58,59,61,62,100,101]. Mackintosh reported that age greater than 14 years, BMI > 45 kg/m^2^, and more significant deformity were associated with failure [100]. Park et al. reported favorable results in hemi-epiphyseal stapling in LOTV [27].

An external fixator can be used to distract through an open proximal tibial physis in LOTV without the need for an osteotomy (Figure 23, 1–5). The correction occurs through the site of deformity and leads to physeal arrest after consolidation and thus prevents recurrence. It requires a stable external fixator, and the surgeon must be aware of the epiphyseal pins’ close proximity to the joint and monitor for infection [102]. De Pablos reported complete correction in 12 of 12 patient with no recurrence [103].

Significant obesity (Figure 24) limits surgical treatment options due to the large exposures needed for internal fixation, risk of internal hardware failure secondary to challenges with mobilization and immobilization, and challenges obtaining stability. Percutaneous or minimally invasive osteotomies with acute or gradual correction with hexapod external fixation is the mainstay of deformity correction in obese adolescents with LOTV who have failed guided growth or are not a candidate due to age or severity of deformities. The hexapod fixator allows for immediate weight bearing and mobility, offers rigid stabilization of the bones, and is very accurate.

Juvenile Blount’s disease was described by Thompson in 1984 in children aged 4–10 with pathologic tibia vara. Little is known about the etiology or natural history. It shares common pathology with early- and late-onset tibia vara. Because the patients are larger than those with ITV and can have significant deformities, the surgical reconstruction strategies are similar to those for LOTV: guided growth and corrective osteotomy with internal or external fixation depending on patient age, weight, and deformities.

## 19. Conclusions

Blount’s disease represents a large spectrum of pathology from infantile (before age 4) to late-onset (after age 4) with a common mechanical pathogenesis leading to medial tibial growth suppression and deformity [104]. The deformity is often multiplanar and includes tibia vara, procurvatum, and tibial torsion. Distal femur and tibia deformities more often exist in the late onset form. Projected limb length discrepancy must be calculated and addressed. Non-operative treatment with bracing in young children has limited efficacy, and surgery is warranted in patients with documented clinical or radiographic progression. Surgical options depend on the patient’s age, extent of physeal involvement, severity and number of deformities, and the psychosocial family dynamics. Despite the unpredictable results and frequent failure, due to the low morbidity and potential to correct deformity or, at least, prevent progression, most authors recommend lateral tension-band hemi-epiphysiodesis as a first line of treatment for most patients regardless of age or weight [10,60,105]. If this fails, then corrective osteotomy is the salvage procedure, preferably before age 4 in ITV with options for acute or gradual correction depending on the patient’s age, size, severity, and complexity of deformity. Completion of lateral tibia and proximal fibula epiphysiodesis is required in late-stage ITV to prevent recurrence. The goals of surgery are to restore normal joint and limb alignment, equalize limb lengths at skeletal maturity, and prevent recurrence, as untreated deformity can progress to knee pain, instability, and degenerative arthritis [46,47,106,107].

## Figures and Tables

**Figure 1 children-08-00566-f001:**
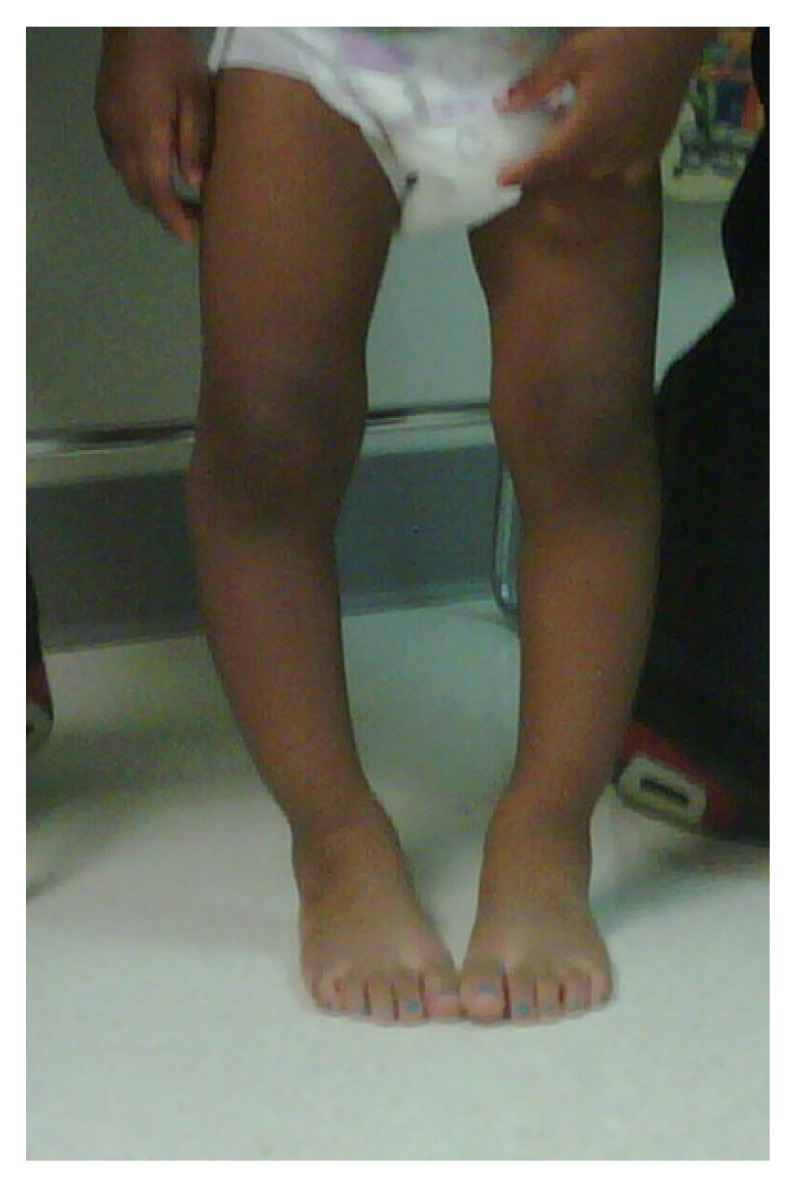
A 20-month-old child with ITV.

**Figure 2 children-08-00566-f002:**
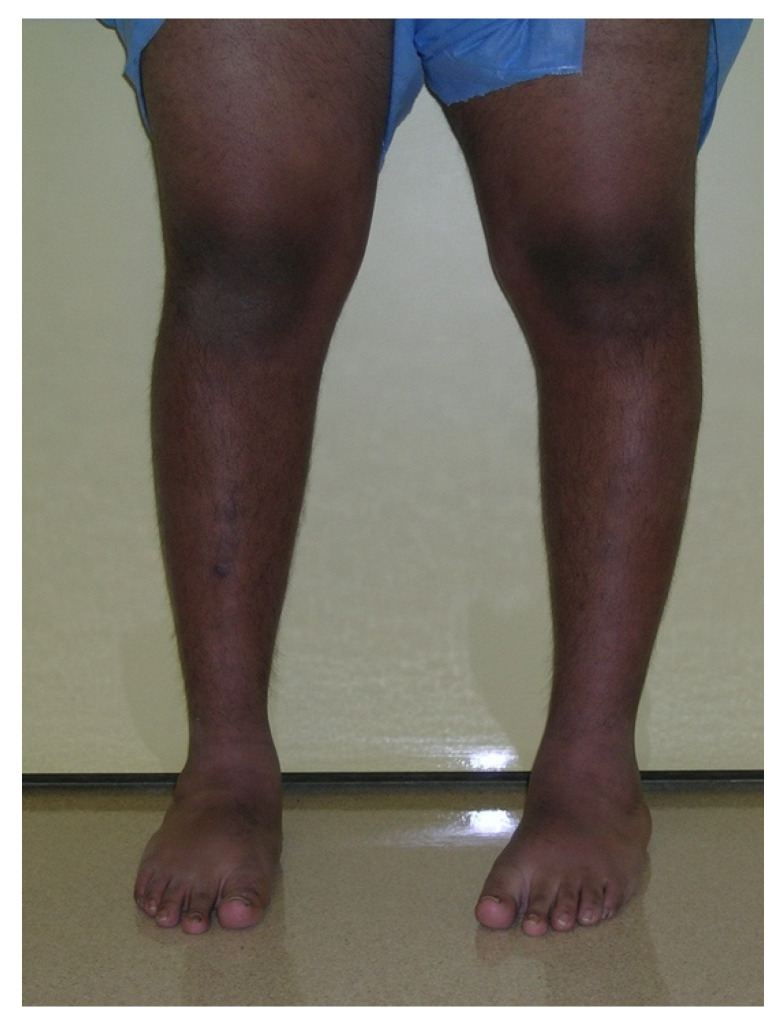
A 14-year-old with left-sided LOTV.

**Figure 3 children-08-00566-f003:**
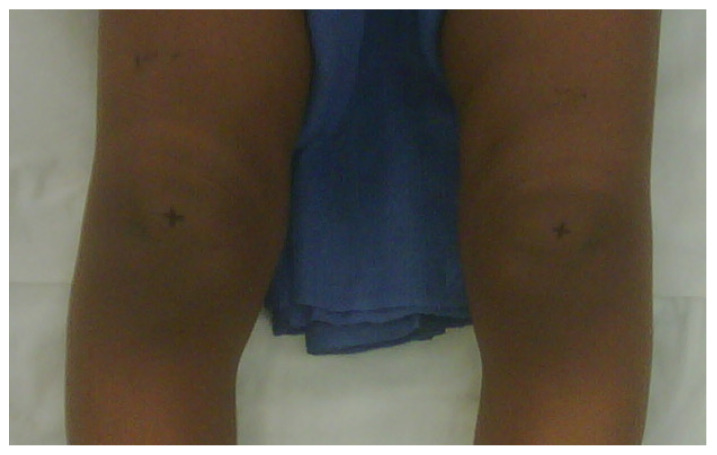
Positive Cover Up test of the child in Figure 1 demonstrating neutral or varus alignment at the lateral knee indicative of ITV.

**Figure 4 children-08-00566-f004:**
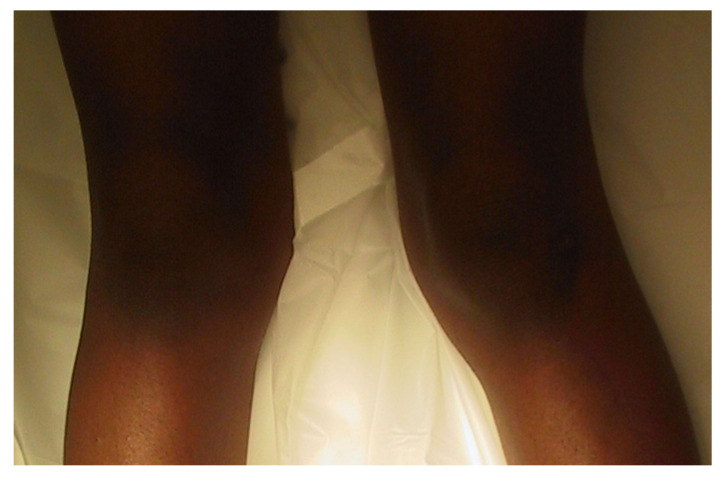
Negative Cover Up test demonstrating valgus alignment at the lateral knee indicative of physiologic genu varus.

**Figure 5 children-08-00566-f005:**
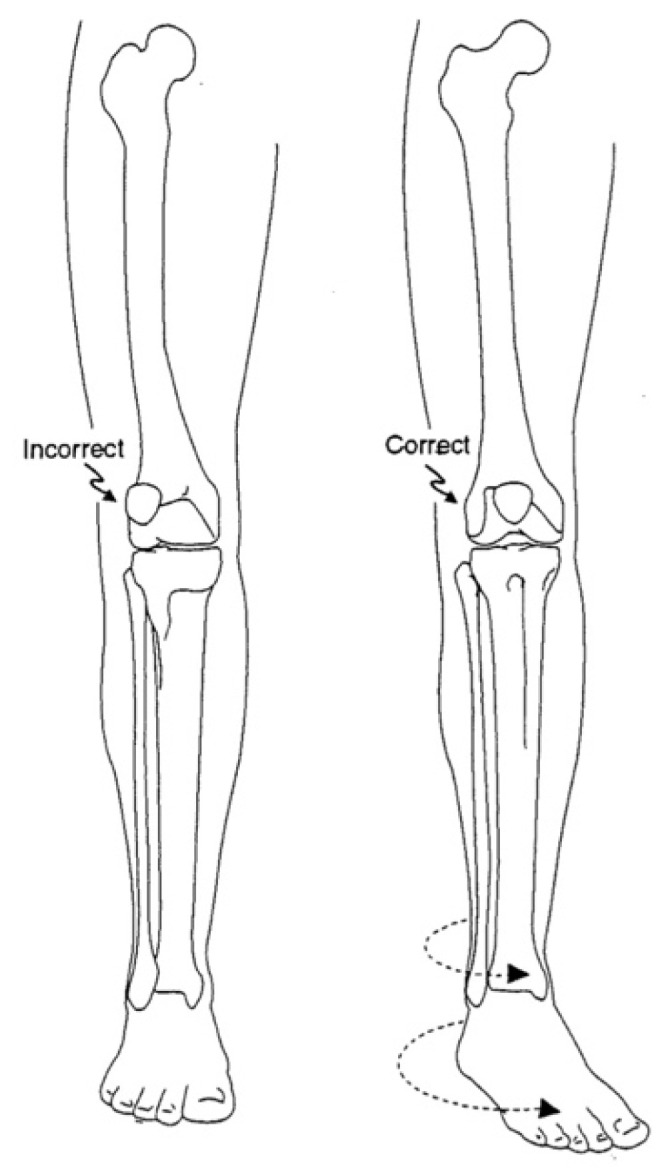
Correct position for AP standing X-ray with patella rotated forward [32].

**Figure 6 children-08-00566-f006:**
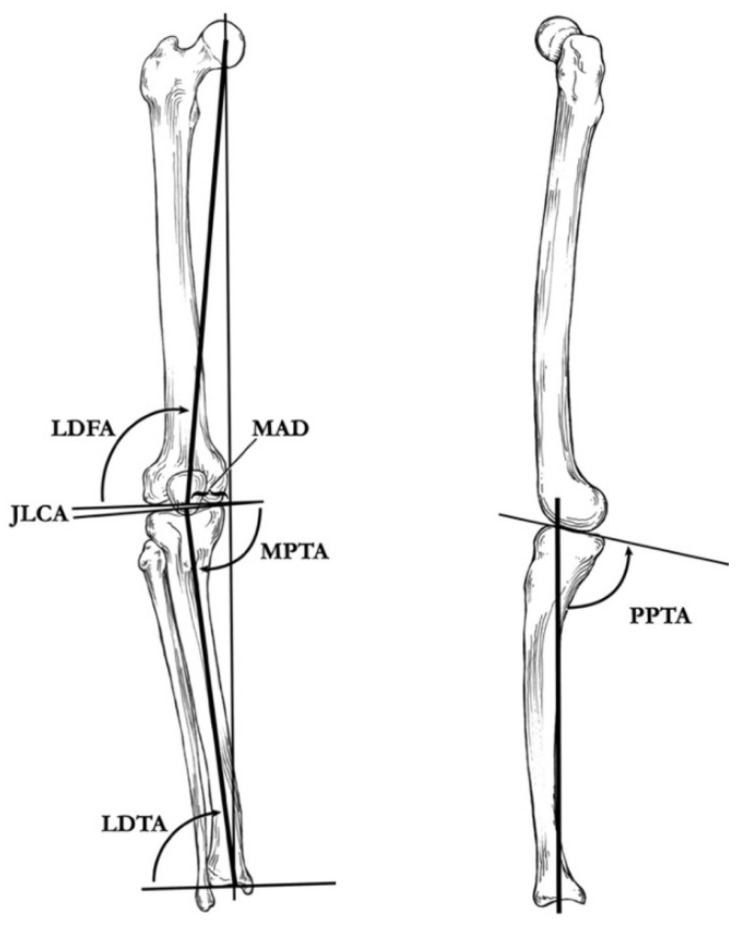
(**Left**): Coronal plane illustration of the lower extremity, demonstrating the mechanical axis deviation (MAD), the lateral distal femoral angle (LDFA), the joint–line congruency angle (JLCA), the medial proximal tibial angle (MPTA), and the lateral distal tibial angle (LDTA). (**Right**): Sagittal plane illustration of the proximal part of the tibia and the distal part of the femur, demonstrating the posterior proximal tibial angle (PPTA) [29].

**Figure 7 children-08-00566-f007:**
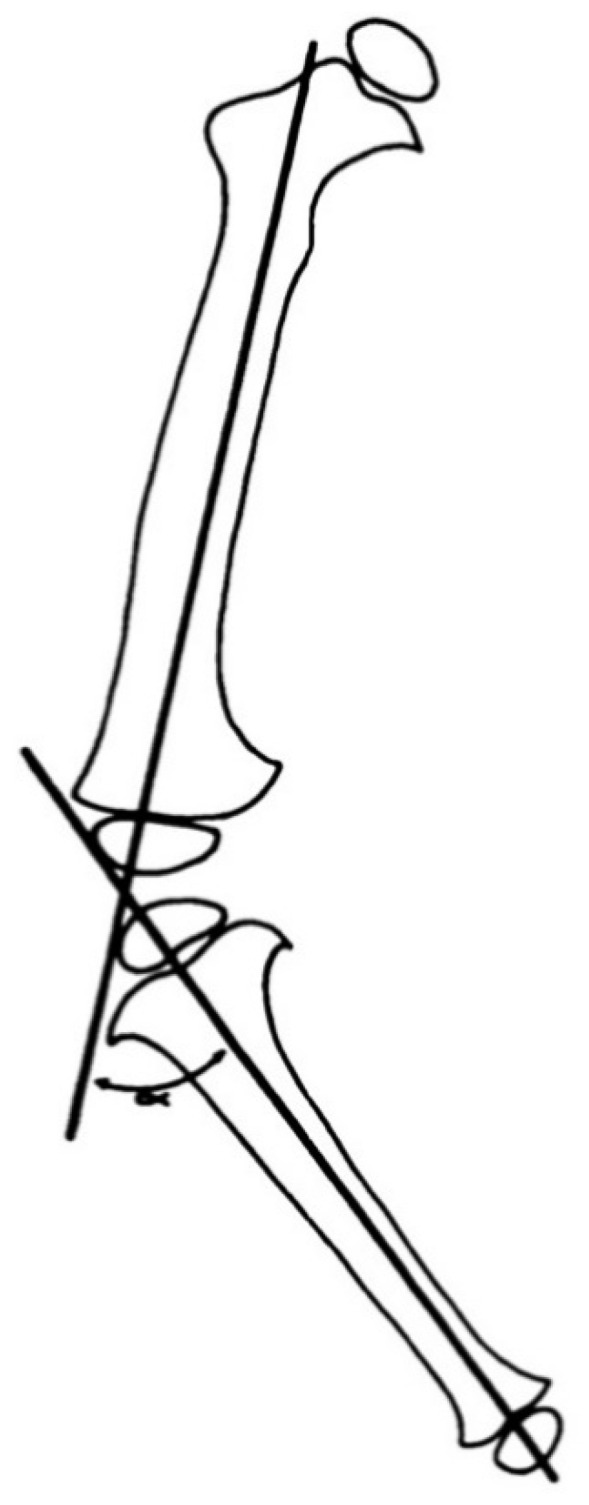
Illustration of the tibiofemoral angle as described by Salenius and Vankka [37].

**Figure 8 children-08-00566-f008:**
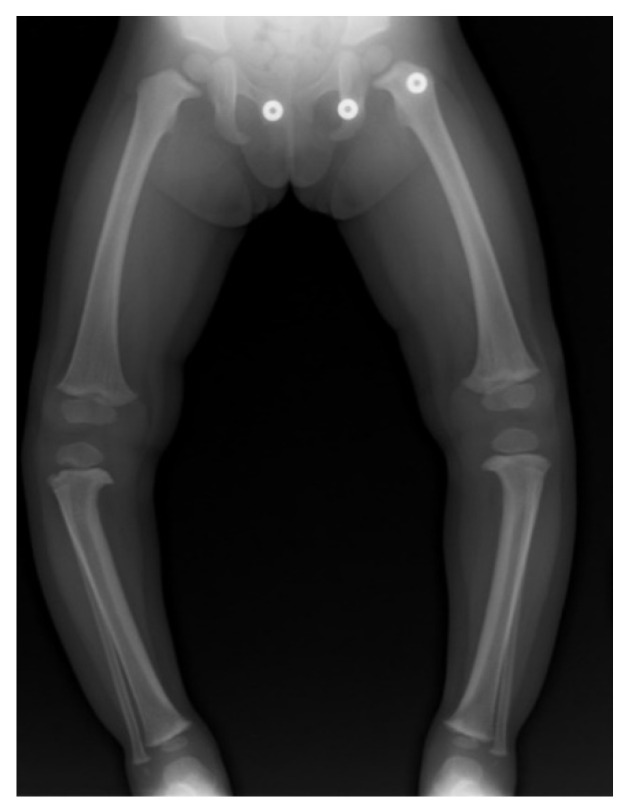
Standing X-ray of the 20-month-old child in Figure 1 with bilateral ITV; MDA right 22°, MDA left 16°.

**Figure 9 children-08-00566-f009:**
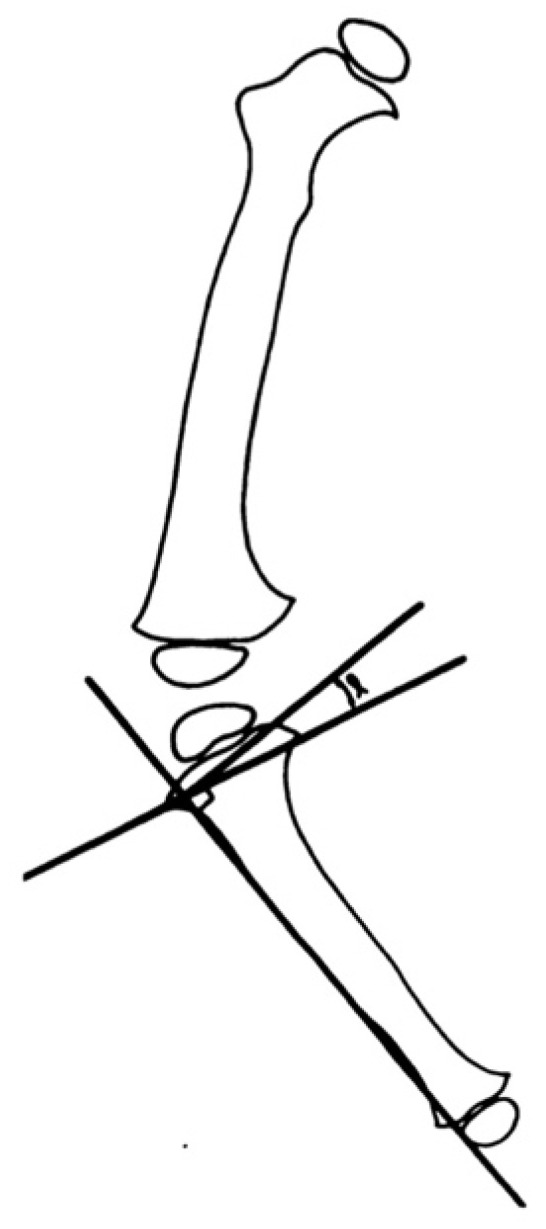
Illustration of the metaphyseal–diaphyseal angle (MDA) of Levine and Drennan. The angle lies between a line drawn through the most distal point on the medial and lateral beaks of the tibia metaphysis and a line perpendicular to the lateral cortex of the tibia [37].

**Figure 10 children-08-00566-f010:**
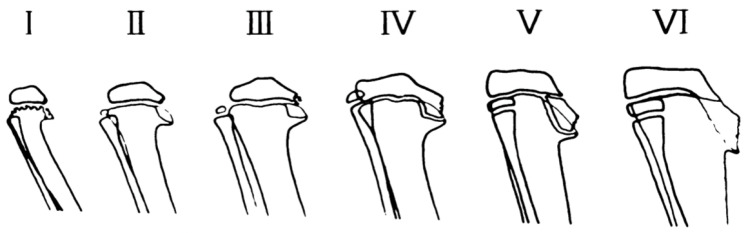
Diagram of radiographic changes seen in the infantile type of tibia vara and their development with increasing age [41].

**Figure 11 children-08-00566-f011:**
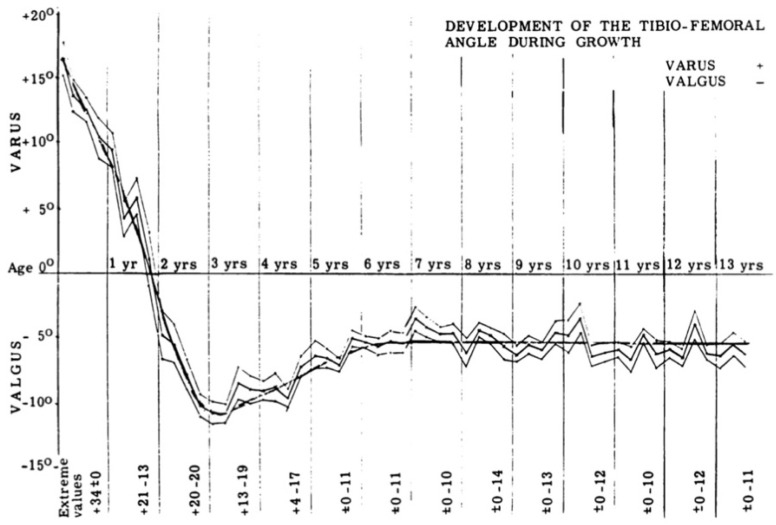
The development of the tibiofemoral angle in children during growth [33].

**Figure 12 children-08-00566-f012:**
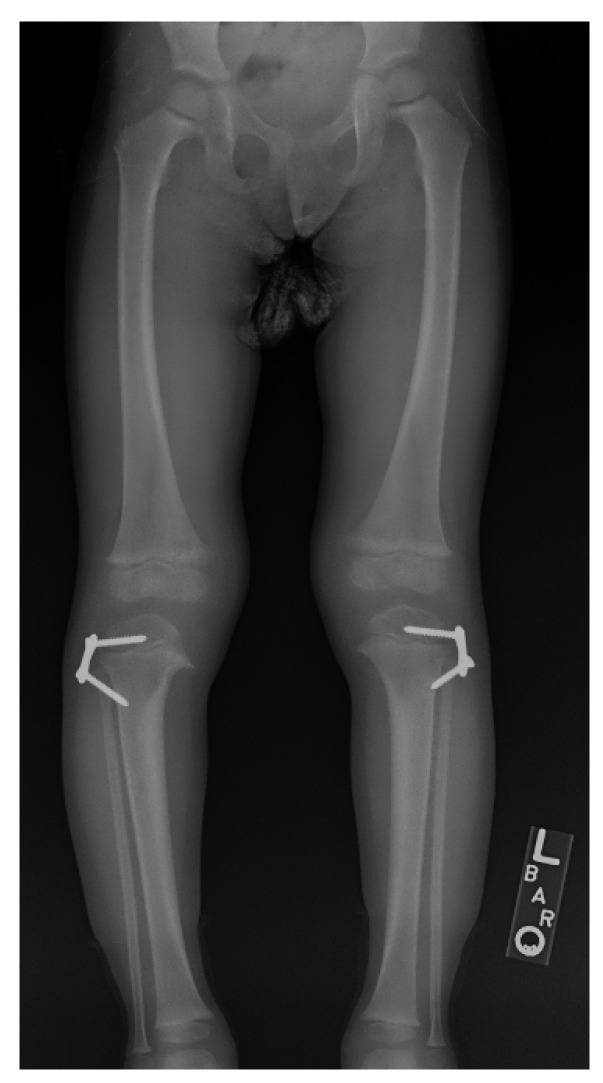
The child in Figure 1 with ITV 10 months after bilateral lateral hemi-epiphysiodesis showing interval correction of varus.

**Figure 13 children-08-00566-f013:**
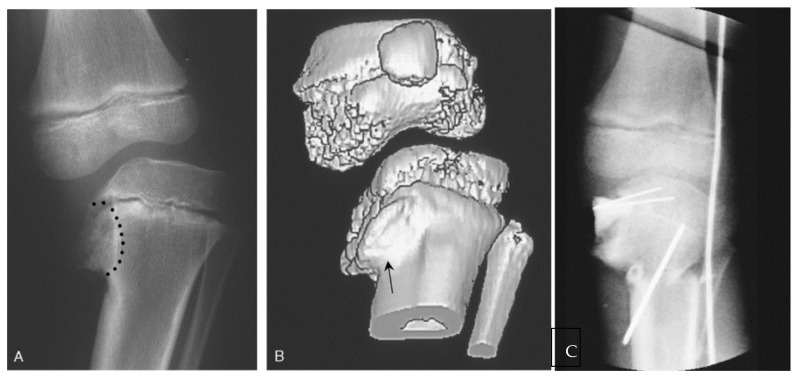
(**A**) A 6-year-old female with untreated stage IV ITV; the metaphyseal bone and Blount lesion to be excised is outlined; (**B**) 3D CT showing extent of abnormal depressed epiphyseal bone anteriorly. The "cleft" on the anterior surface of the metaphysis (arrow) is where the initial resection for epiphysiolysis begins. (**C**) After interposition of cranioplast at the site of resection and stabilization of a closing wedge tibial osteotomy. The bovie cord defines the lateral mechanical axis [65].

**Figure 14 children-08-00566-f014:**
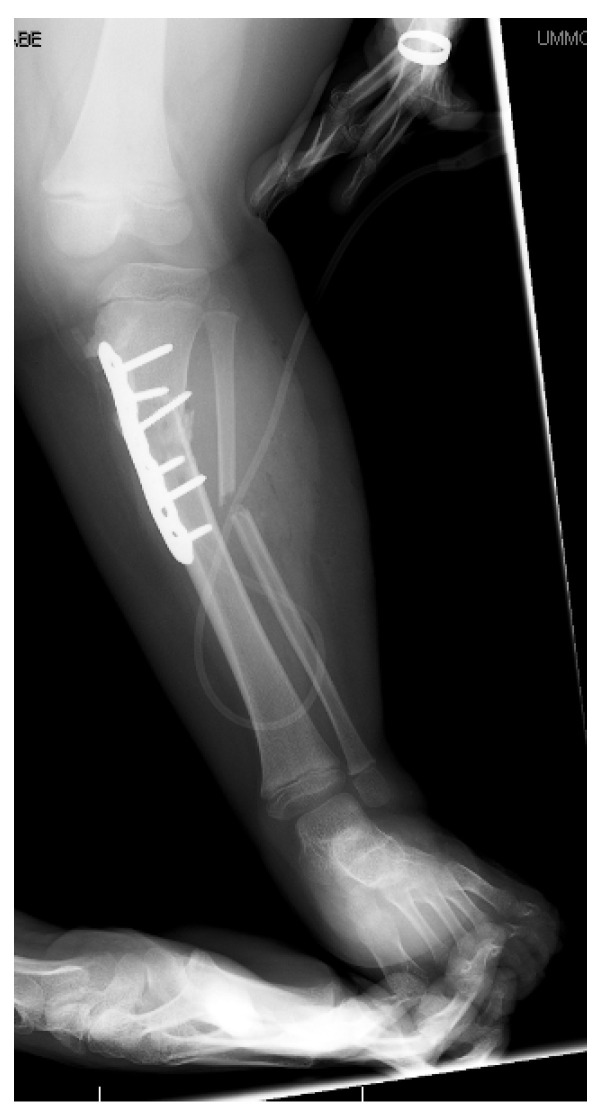
A 4-year-old with II/III ITV with acute over-correction into valgus.

**Figure 15 children-08-00566-f015:**
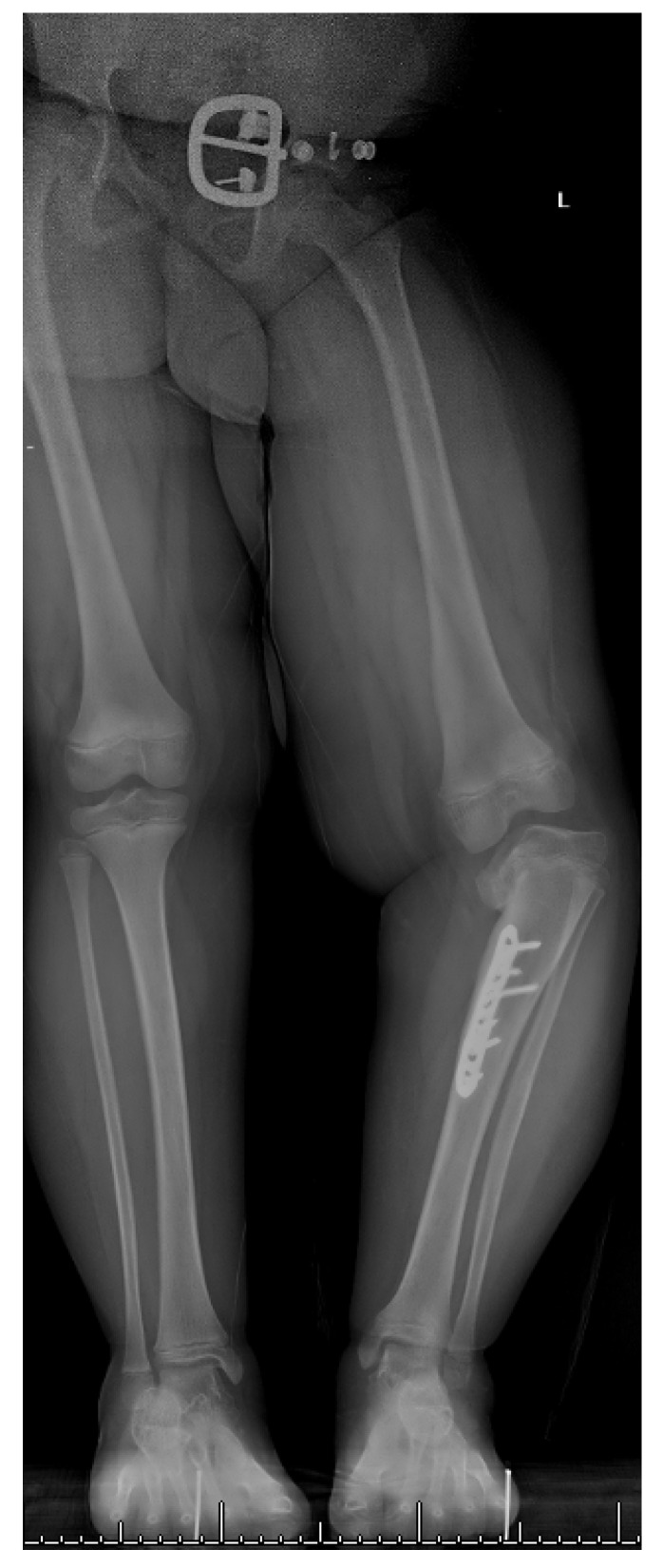
A 7-year-old child who failed acute valgus-producing osteotomy due to medial physeal bar formation; note the medial tibial plateau depression and limb length discrepancy.

**Figure 16 children-08-00566-f016:**
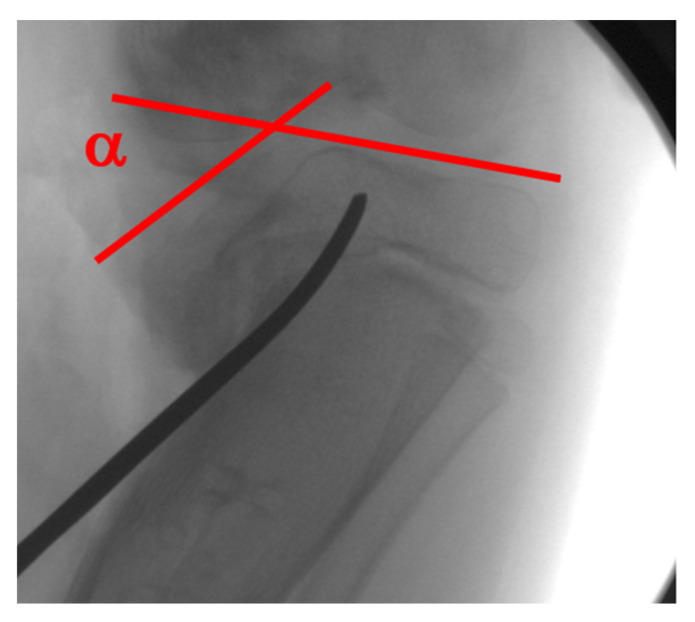
The medial tibial plateau depression angle (α) is measured between a tangent to the lateral plateau and medial plateau; the osteotomy aims towards the intercondylar notch.

**Figure 17 children-08-00566-f017:**
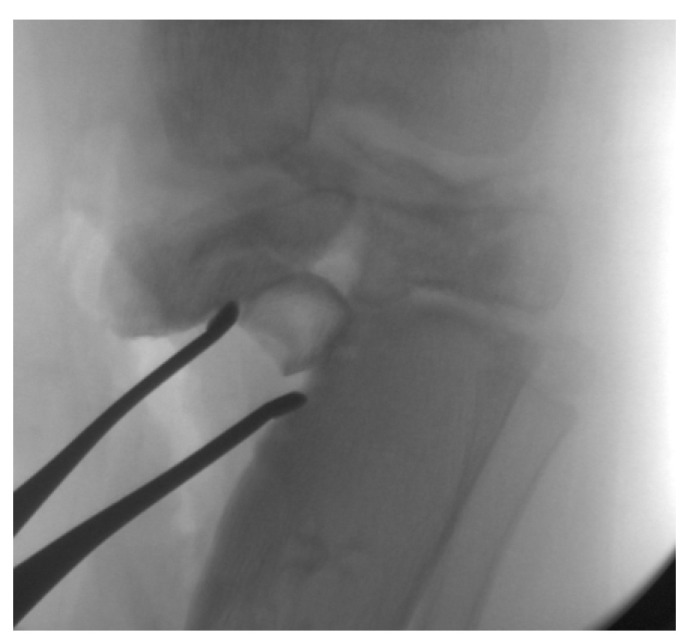
The osteotomy is complete and held in a corrected position to restore the medial joint line; the iliac crest allograft is impacted in the opening wedge.

**Figure 18 children-08-00566-f018:**
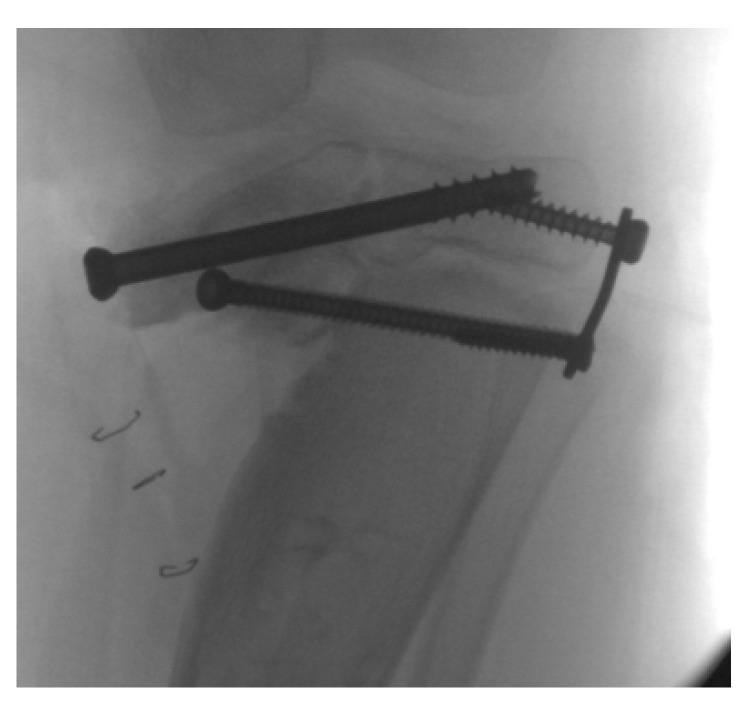
The osteotomy and allograft are stabilized with screws; the proximal tibia epiphysiodesis is completed by placing a lateral tension band to prevent recurrence; a percutaneous ablative epiphysiodesis of the proximal fibula was performed.

**Figure 19 children-08-00566-f019:**
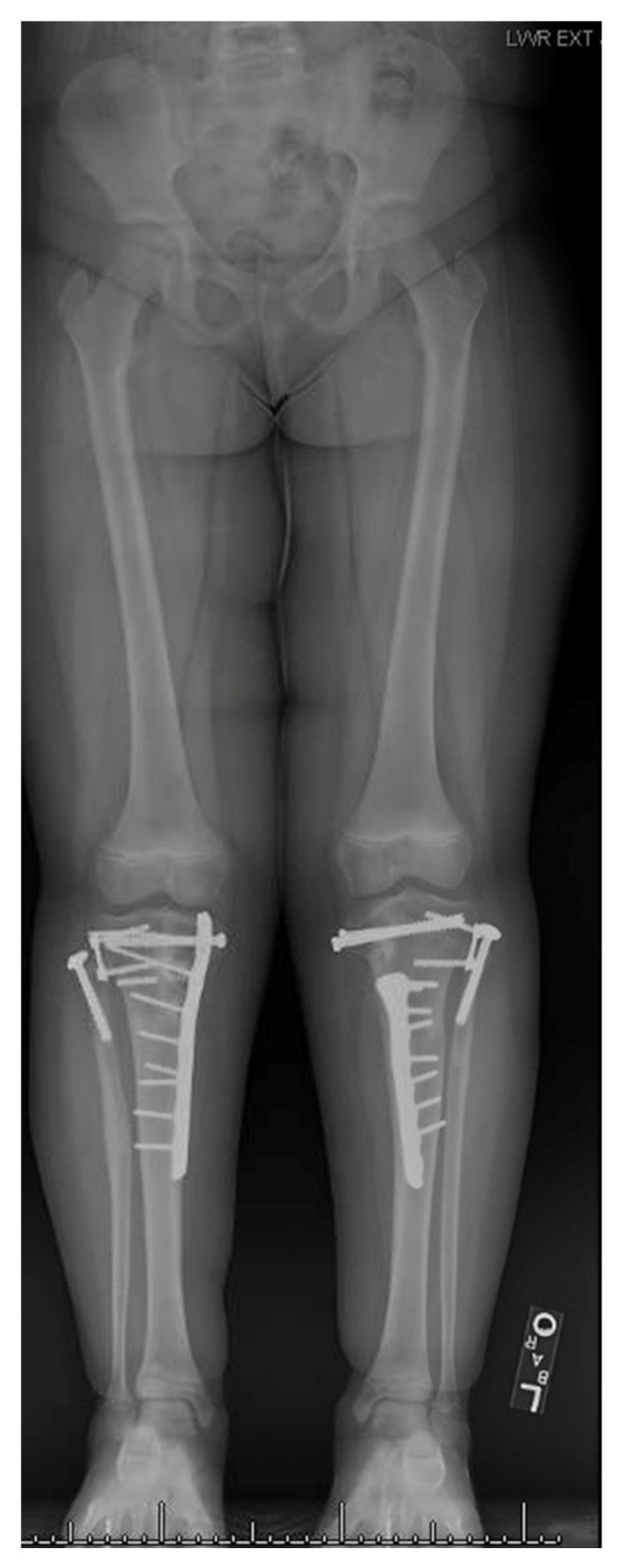
A 7-year-old with bilateral advanced ITV underwent staged acute correction and growth plate stoppage.

**Figure 20 children-08-00566-f020:**
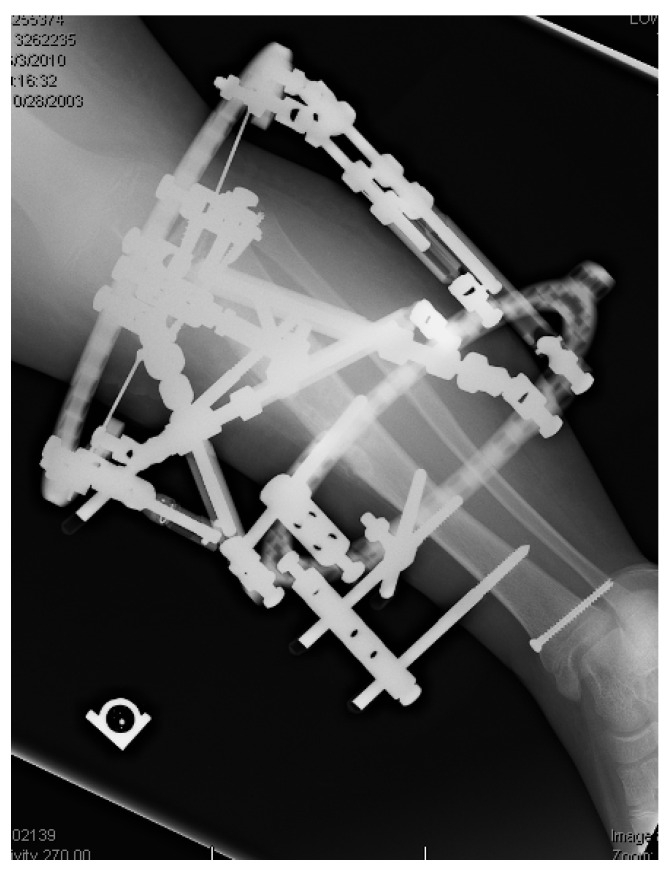
The patient in Figure 16, Figure 17 and Figure 18; a hexapod was placed to correct tibia varus, internal torsion, and procurvatum; 3 cm of length was gained due to the growth plate closures.

**Figure 21 children-08-00566-f021:**
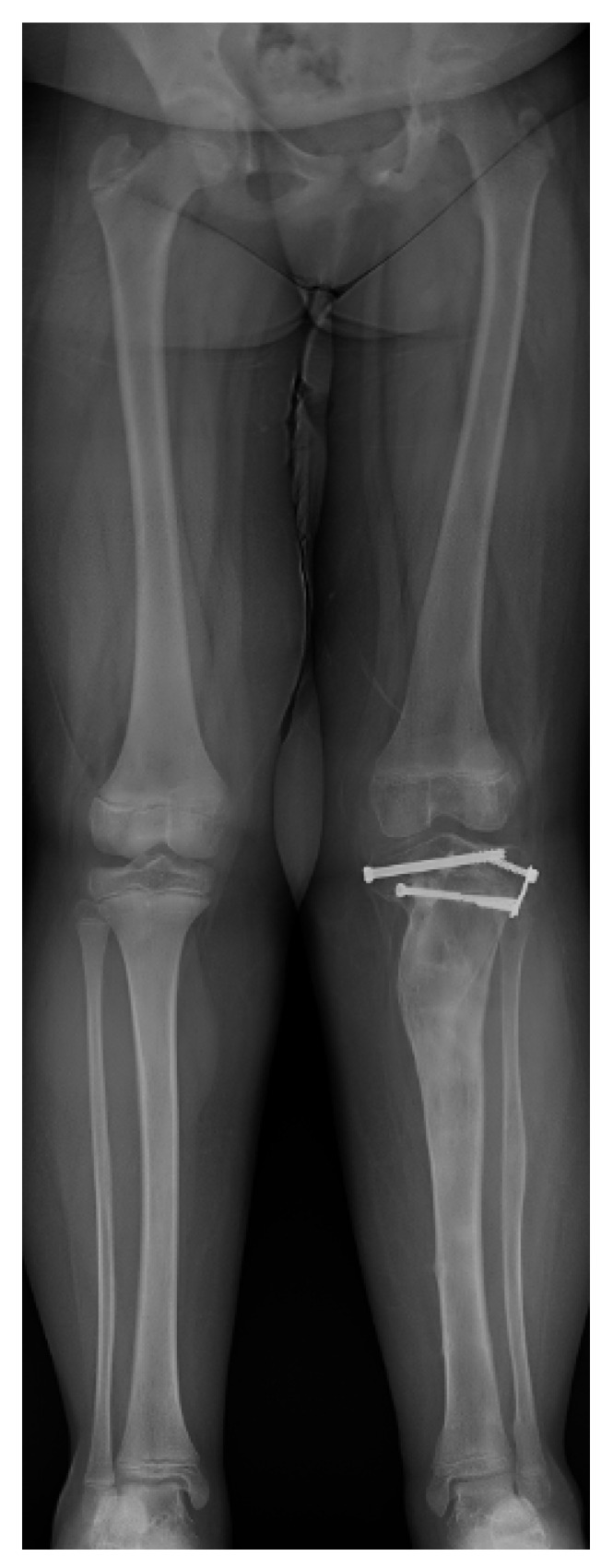
The patient in Figure 15, Figure 16, Figure 17 and Figure 18 6 months after frame removal showing maintained alignment and length.

**Figure 22 children-08-00566-f022:**
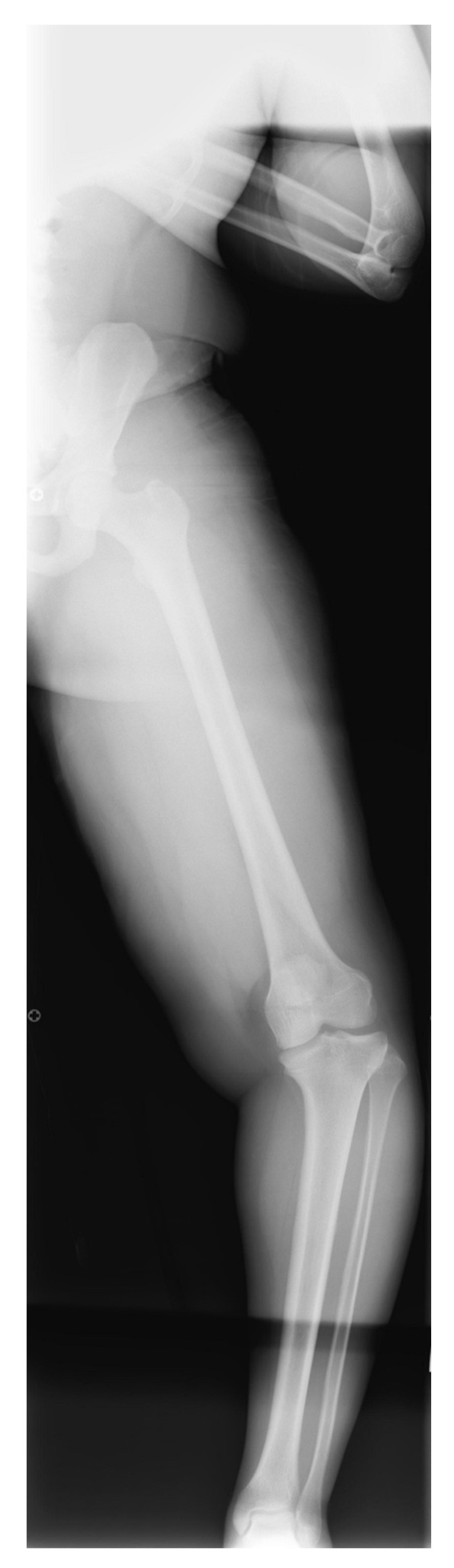
A 16-year-old adolescent with unilateral LOTV; he has distal femur varus, proximal tibia varus, and distal tibia valgus.

**Figure 23 children-08-00566-f023:**
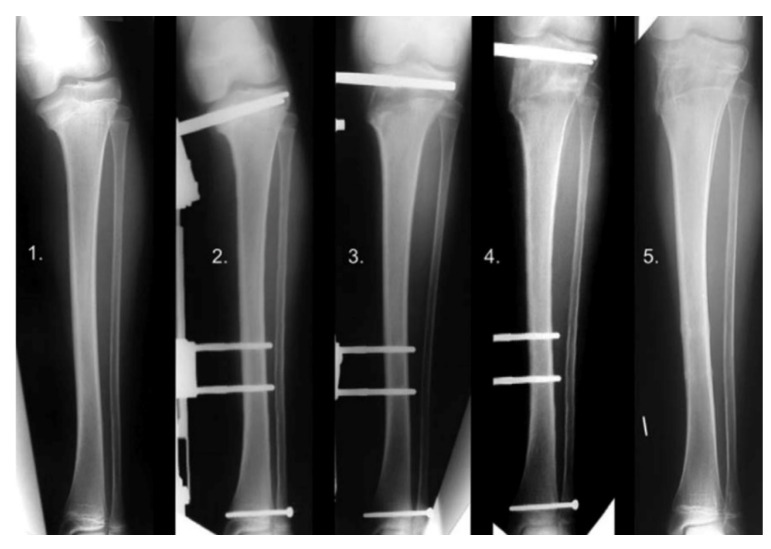
Physeal distraction in LOTV. X-ray series of a case of LOTV gradually corrected by means of physeal distraction: (**1**) Pre-operative, (**2**) immediate post-operative, (**3**) 22 days post-operative angular correction completed, (**4**) 2 months post-operative after 2.5 cm lengthening to compensate a discrepancy, (**5**) 4 months post-operative consolidation obtained [96].

**Figure 24 children-08-00566-f024:**
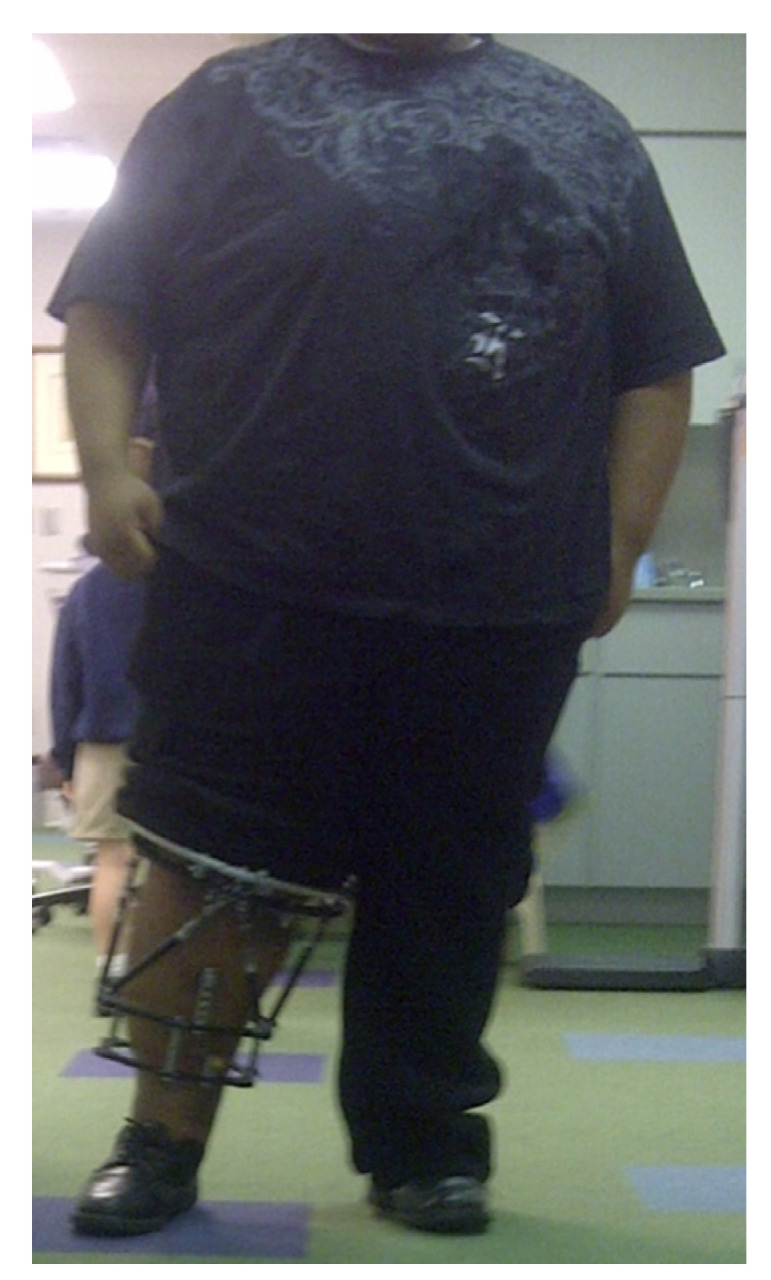
A 400-pound (181 kg) adolescent with a hexapod fixator stabilizing gradual correction of LOTV.

## Data Availability

This is a literature review article; no data was accumulated.

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
