# Peer review of "Deformity Reconstruction Surgery for Blount’s Disease"

_children, 2021, doi:10.3390/children8070566_

Round 1
Reviewer 1 Report
Dear Author,
I have some questions and recommendations:
- Since there are different types of review article, please indicate the type of the present review (for example narrative, systematic, critical, overview etc.). Also indicate the type in page 1/26 (abstract, line 25) and in page 22/26 (Data availability, line 582).
- Present manuscript is a literature review. Although, I cannot found about “Baboo osteotomy”? Pros and cons? Siregar (2010) has described a technique of longitudinal osteotomies.
- Advantages of CT 3D and 3D printing for pre-operative planning in Blount disease? It seems that using 3D printed guides allow for accurate correction of the deformity (Gomez-Dalono et al., 2020)
- Figures: Very nice and illustrative figures. Only two points for correction, figure 13 C) is not clear (if it is possible, please replace) and also “C” is not indicated in the figure. Also Figure 20 is not clear (please replace if it is possible).
- References: Nothing to add.
- A more carful editing of the manuscript is indicated to avoid minor grammar mistakes.
Kind regards,
Angelo V. Vasiliadis, MD, PhD

Author Response
1) 'literature' added to both lines
2) I read the article on Bamboo osteotomy and disagree with the author that it is minimally invasive. As well, the idea of using an oscillating saw to make multiple longitudinal bone cuts is not something I would consider due to the heat generated and multiple potentials for iatrogenic injury each time the saw is passed. I found 8 references to this article--those in English referenced this single article with a single patient. The author has not reported follow-up of this one case or presented additional ones. I do not think it is worthy to include.
3) I included the reference to the 3d printing in the section of LOTV
4) I moved the "C" to the figure indicated; figure 20 with the hexapod is, unfortunately, what xrays look like with hexapod fixators--a lot of metal and very little bone visible
5) I re-edited and found additional errors. thank you.
Reviewer 2 Report
Thank you for this interesting overview on Blount's disease and the treatment options.
Some suggestions and questions:
- Please adjust the title to cover the content of the article
- What is a positive and a negative test line 113 and further and both figure 3 and 4. What is the alignment without the cover?
- Radiographic Imaging line 130 how does a radiograf demonstrate joint laxity?
- Diagnosis line 166 why as only the radiograph described for diagnostic purposes? Please elaborate on other diagnostic options and there values
- The natural history paragraph can be eliminated
- Please address the APA in lines 223-225
- Line 323 osteotomy options please add W-shaped osteotomy and elaborate on the technique
- Line 380 acute versus gradual is more appropriate as a final remark
- Paragraph 14 hemiplateau elevation line 385 and paragraph 16 line 438 please report on the outcome of these techniques
- Lines 523 to 533 please report on the outcome of both the guided growth and the distraction of the proximal tibial physis. What are the pros and cons?
- Can you please clarify the reason for lines 551 to 556
Author Response
1) this title was assigned; the "review" above the title should indicate the type of article
2) figure headings and test description in the text were modified
3) text modified
4) a clinical suspicion of ITV is confirmed with radiographs;
5) I added additional verbiage to this section
6) I think it's appropriate to include the natural history in this review
7) I'm not sure what you mean by the APA in lines...
8) W/M added
9) I'm not sure what you mean by 'acute vs gradual...
10) #14 verbiage ammended--results are reported in #15; #16 referrences added
11) verbiage changed and references added
12) references cited
13) the introduction tells of a 3rd category: juvenile. although i mentioned i would discuss ITV and LOTV i thought it worth mentioning that juvenile blount's behaves like both
Reviewer 3 Report
This is invited review manuscript on Blount disease and extensive review of surgical treatment options. It is well and logically organized. However issue editor is partner of the author of this manuscript. And it is obvious that there are more citations of the group author works with. This is not inappropriate but it is clearly visible.
In the section No 14 i there is detailed description of hemi-plateau elevation as surgical option for selected stage of Blount disease. However this option is not mentioned at all in introduction section and in conclusion. It is suggested to correct this omission.
Specific comment (maybe trivial) is referring to Figure 1 where is photo of child in high socks covering significant part of lower leg. It is suggested that author should find better photo to illustrate his/her point.
In the line no.141 there is spelling of the name as Salenias. it should be Salenius. The same thing in the line 161.
There are 105 references listed at the end of the manuscript. Superficial overview has shown duplicate of references, e.g. references no. 24 and no. 25 are duplicate, also references no. 7 and no. 30, ; references no. 28 and no. 92. reference 32 is incomplete. It is suggested to go more carefully through reference list, as many readers are using them for further reading.
As this review manuscript should be comprehensive and updated it is suggested to include recent references e.g. Mare PH et al. JPO 2021. who analysed 64 limbs in 48 patients, and also Baraka MH et al. JCO 2021. who analysed 21 knees in 19 patients. Long-term (65 years) follow-up is recently published in Acta Orthopaedica.
Author Response
1) thank you for this comment; "hemi-plateau elevation" is a type of osteotomy so i added the word "osteotomy" to the section 14 and where appropriate in the remainder of the manuscript.
2) i found / replaced this with the clinical picture of a child whose x-rays are used later in the article
3) corrected to Salenius
4) thanks; corrected
5) mare and Baraka articles added and referenced
Round 2
Reviewer 2 Report
Adequate changes and answers
- font size 230-232 (APA)